# Co-Networks Poly(hydroxyalkanoates)-Terpenes to Enhance Antibacterial Properties

**DOI:** 10.3390/bioengineering7010013

**Published:** 2020-01-21

**Authors:** Tina Modjinou, Davy Louis Versace, Samir Abbad Andaloussi, Valérie Langlois, Estelle Renard

**Affiliations:** 1Institut de Chimie et des Matériaux de Paris Est, Univ Paris Est Creteil, F-94320 Thiais, France; modjinou@icmpe.cnrs.fr (T.M.); versace@icmpe.cnrs.fr (D.L.V.); renard@icmpe.cnrs.fr (E.R.); 2Institut de Chimie et des Matériaux de Paris Est, Univ Paris Est Creteil, F-94010 Créteil cedex, France; abbad@u-pec.fr

**Keywords:** poly(hydroxyalkanoate)s, linalool, thiol-ene, photochemistry, antibacterial properties

## Abstract

Biocompatible and biodegradable bacterial polyesters, poly(hydroxyalkanoates) (PHAs), were combined with linalool, a well-known monoterpene, extracted from spice plants to design novel antibacterial materials. Their chemical association by a photo-induced thiol-ene reaction provided materials having both high mechanical resistance and flexibility. The influence of the nature of the crosslinking agent and the weight ratio of linalool on the thermo-mechanical performances were carefully evaluated. The elongation at break increases from 7% for the native PHA to 40% for PHA–linalool co-networks using a tetrafunctional cross-linking agent. The materials highlighted tremendous anti-adherence properties against *Escherichia coli* and *Staphylococcus aureus* by increasing linalool ratios. A significant decrease in antibacterial adhesion of 63% and 82% was observed for *E. coli* and *S. aureus*, respectively.

## 1. Introduction

Nosocomial infections constitute a serious healthcare concern. Each year, more than 1.4 million cases are diagnosed in the world, unfortunately leading to deaths and to an incredible increase in healthcare costs [1,2,3,4,5]. Some studies [4,5] provide estimates of the number of health care-associated infections (HAIs) annually observed in the US: Approximately 2 million patients suffer from HAIs, and nearly 90,000 are estimated to die. In 2050, the projected mortality rate linked to HAIs could reach 10 million annually, which is higher than the projected rate for cancer. These hospital-contracted infections are due to the development of pathogenic microorganisms, such as *Escherichia coli* (*E. coli*), *Staphylococcus aureus* (*S. aureus*) or *Pseudomonas aeruginosa (P. aeruginosa)*. Moreover, with the excessive use of antibiotics, the development of multi-resistant bacterial strains is constantly increasing, and the phenomenon is not going to stop. Thus, to prevent these infections, investigations are dedicated to design well-performing antibacterial materials. To date, the antibacterial materials could be classified into two types. On one hand, passive materials, which prevents bacterial adhesion and can be obtained by the use of poly(ethylene glycol) [6], polysaccharides [7], N-acetylpiperazine or fluorinated compounds [8,9]. On the other hand, active or biocide materials can be prepared via the incorporation of metal nanoparticles which are well-known for their antibacterial activities, such as silver [10] and copper [11], by using polymers having ammonium or sulfonium salts [12] and organic compounds such as N-halamines [13], phenols [14,15] and aromatic acids [16,17]. Essential oils constitute a source of terpenes that are natural molecules, which are known as antibacterial and antimicrobial compounds [18,19,20,21,22]. Among them, linalool [20,21,23,24,25], the main monoterpene component from lavender oil, has been classified by the Food and Drug Administration as a Generally Recognized As Safe (GRAS) compound [26].

The objective is to combine linalool with a bio-based polymer to develop novel antibacterial materials with a significant decrease in the carbon fossil percent that is consistent with socio-economic issues nowadays [27,28]. Among the bio-based polymers, poly(hydroxyalkanoate)s (PHAs) are aliphatic polyesters produced by bioconversion as intracellular nutriment storage materials inside bacteria [29,30,31]. Two types of PHAs with various physical properties can be distinguished according to the length of the side chains: Short chain length (scl-PHAs), possessing alkyl side chains having up to two carbon atoms, and medium chain length PHAs, or mcl-PHAs, having at least three carbon atoms on their side chains. Scl-PHAs are semi-crystalline, rigid and brittle, and on the opposite, medium chain length (mcl-PHAs) present elastomeric properties. Finally, according to their promising properties, such as biodegradability and biocompatibility, they can be used in biomedical applications [32].

Terpenes and poly(3-hydroxybutyrate-co-3-hydroxyvalerate) (PHBHV) are used to elaborate co-networks by a photo-induced thiol-ene reaction in the presence of tri- or tetra-thiol. PHBHV, chosen in the present study, is the most common natural scl-PHA produced from renewable sources by a great variety of microorganisms. The thiol-ene reaction is now widely used [33] due to many advantages, such as high homogeneity [34,35], low stress and shrinkage in the materials [36], as well as high conversions and no sensitivity to oxygen or water [33,37]. Up to date, only a few photoactivated thiol-ene reactions have been developed for the design of materials from vegetable oils [38,39,40,41,42], terpenes [43,44,45] or plant-derived compounds [46,47]. A preliminary functionalization of poly(hydroxybutyrate-co-hydroxyvalerate) PHBHV is required to introduce the unsaturated terminal groups on PHBHV (Scheme 1). The α,ω-dihydroxylic PHBHV oligomers are first synthesized by a transesterification in presence of ethyleneglycol, which further reacted with allyl isocyanate to obtain α,ω-diallylic PHBHV. The influence of the nature and the content of the crosslinking agent based on multifunctional thiol (TriSH or TetraSH) on the thermo-mechanical performances of the co-networks were carefully evaluated. The anti-bacterial activity against two bacterial strains (*E. coli* and *S. aureus*) of the co-networks was investigated through adhesion tests.

## 2. Materials and Methods 

### 2.1. Materials

Poly(3-hydroxybutyrate-co-3-hydroxyvalerate), PHBHV, with 12 mol % of hydroxyvalerate was purchased from Goodfellow. PHBHV was first purified by dissolution in dichloromethane overnight (10% w/v) and precipitated in petroleum ether for removing citric ester used as plasticizer. 2,2-dimethyl-2-phenylacetophenone (DMPA) was provided by BASF. Anhydrous ethylene glycol (>99.8%), allyl isocyanate (98%), linalool (97%), dibutyltin dilaurate (>95%) (DBTL), trimethylpropane tris-(3-mercaptopropionate (>95%) (TriSH) and pentaerythritol tetrakis (3-mercaptopropionate) > 95% (TetraSH) were obtained from Sigma-Aldrich (Saint louis, MO, USA). Dichloromethane (>99.8%) and chloroform (SDS, analytical grade) were dried over CaCl_2_, petroleum ether (>98%), acetone, n-pentane, and ethyl acetate were used as received.

### 2.2. Synthesis of α,ω-Dihydroxylic PHBHV Oligomers under Microwave Irradiation

The microwave-assisted transesterification was carried out on an Anton-Paar monomode 300 microwave reactor with a magnetron with a 2.5 GHz frequency powered by a 900 W power generator as microwave source. PHBHV oligomers were obtained by transesterification of PHBHV (Mn = 90,000 g/mol) with ethylene glycol. Typically, 1 g of PHBHV (1.11 × 10^−5^ mol) was dissolved in 10 mL of anhydrous CHCl_3_ in a 30 mL microwave transparent glass vial under stirring. Then, 630 µL of anhydrous ethylene glycol (9.28 × 10^−3^ mol) and 1.24 mL of DBTL (2.09 × 10^−3^ mol) (for a molar mass of 1000 g/mol) were introduced and the mixture was stirred for 5 min. A calibrated temperature control mode was used to perform the microwave assisted reaction. This mode consisted in reaching a set temperature ramp of 140 °C in 3 min, and then this temperature was maintained during 60 min. The reaction was terminated by cooling with compressed air to 50 °C. The internal reaction temperature was monitored and controlled with a fiber-optic sensor (Ruby thermometer) immersed in the mixture. The mixture was then precipitated in 150 mL of pentane and filtered. White powder obtained was then solubilized in dichloromethane and precipitated in a 10-fold excess of diethyl ether to eliminate DBTL. The product was then dried under vacuum at 60 °C overnight (yield 80%).

### 2.3. Synthesis of α,ω-Diallylic PHBHV Oligomers

The α,ω-dihydroxylic PHBHV oligomers (0.8 g) were dissolved in 8 mL of anhydrous chloroform. Under argon bubbling, 22 µL of allyl isocyanate and 10 µL of DBTL were added. The mixture was stirred at 80 °C (the vessel was hermetically closed) during 4.5 h and then precipitated drop-wise in 100 mL of n-pentane. After filtration and drying under vacuum at 60 °C overnight, the α,ω-diallylic PHBHV oligomers were obtained with a yield of 90%.

### 2.4. Synthesis of PHBHV–Linalool (PHBHV-L) Co-networks

As an example, the synthesis of the 70–30 wt% PHBHV-L co-networks is described here (Scheme 1). The α,ω-diallylic PHBHV oligomers (210 mg, 2.10 × 10^−4^ mol) were dissolved into 1 mL of DMPA solution in acetone (3% g/g) and the minimum volume of dichloromethane. Linalool (90 mg, 5.83 × 10^−4^ mol) and trithiol (237 mg, 5.95 × 10^−4^ mol) or Tetra-SH (356.7 mg, 7.93 × 10^−4^ mol) were added. To the previous PHBHV-derived solution, reagents were introduced with respect to the stoichiometry of functions (n_C=Ctot_ = n_SH_). The final mixture was introduced into a silicon mold (2.5 × 4 cm) and was irradiated under UV light for 10 min (sample placed at 14 cm of the end of the guide of a Hamamatsu Lightning cure LC8 (L8251) equipped with a mercury-xenon lamp (200 W)).

### 2.5. Characterization

Monomers Characterization: ^1^H NMR (400 MHz) spectra were recorded on a Bruker AV 400MHz in CDCl_3_ at room temperature or in DMSO-d6 at 70 °C. Size Exclusion Chromatography (SEC) analysis were performed on an apparatus equipped with Shimadzu-LC-10A pump connected to two PL-gel Polymer Laboratories (Mixte-C of 5 μm) columns mounted in series on a Wyatt Technology Optilab Rex refractometer interferometer used as detector. Chloroform was used as eluent with a flow of 1 mL/min at 25 °C. Polysciences Polystyrenes standards were used to the calibration. The samples were analyzed at a concentration of 10 mg/mL.

Spectroscopic Analyses: ATR-FTIR spectra were recorded on a Bruker Tensor 27 spectrometer equipped with an ATR apparatus. Water contact measurements were performed with a drop shape analysis system Krûss Easy Drop contact angle measuring system apparatus controlled by DSA software. Photo-polymerization kinetics were followed by real-time Fourier transform infra-red spectroscopy (RT-FTIR) using a Jasco 4700 Instrument. As an example in the case of the 30–70 PHBHV–Linalool co-network, 80 μL of a mixture of PHBHV (30 mg, 2.6 × 10^−5^ mol) dissolved in 100 μL of dichloromethane, linalool (70 mg, 4.5 × 10^−4^ mol), cross-linking agent (109.3 mg, 2.7 × 10^−4^ mol (triSH) or 99.6 mg, 2.0 × 10^−4^ mol (tetraSH)) and the photo-initiator (6.3 mg or 6.0 mg, 3 wt% of the monomers) were applied to BaF_2_ chips by means of a calibrated wire-wound applicator and exposed for few seconds to both the IR beam (which analyzes in situ the extent of the photo-reaction) and the UV beam (which starts the photoreaction). The mixture was irradiated at room temperature using a Hamamatsu Lightning Cure LC8 equipped with a Hg-Xe lamp (200 W) coupled with a flexible light guide. The sample and the end of the light guide was separated by a 14 cm distance. The photo-polymerization was monitored by following the disappearance of the C=C bond of PHBHV at 1647 cm^−1^ and the SH functions at 2560 cm^−1^.

Water Contact Determination: Water contact angle measurements were performed using the Drop Shape Analysis system on a Krüss goniometer.

Water Uptake Determination: Prior to water uptake determination, the sample was dried under vacuum. They were placed in a conditioning closed chamber over a saturated solution of NaCl to obtain a water-saturated atmosphere of 75% at room temperature. The percentage uptake of moisture into the material was measured by coulometric method at t = 0 s and t = 24 h on a 831 KF Coulometer equipped with a 728 stirrer and a 860 KF Thermoprep.

Soluble Extractable Determination: The samples were treated with 15 mL of dichloromethane (to remove unreacted PHBHV monomer) stirred in a flask at 35 °C. The solvent fraction was renewed every 15 min and the process was performed 5 times. The same procedure was repeated in ethyl acetate (to remove unreacted linalool monomer). The rinsed films were weighted after the solvent evaporation under vacuum at 60 °C during 4 h.

Thermal and Mechanical Analyses: Differential scanning calorimetry (DSC) analysis were performed in a Perkin Elmer Diamond apparatus with the following program: a ramp from −60 to 170 °C at 20 °C·min^−1^, an isothermal during 5 min, a cooling at 200 °C·min^−1^ until −60 °C and a heating until 170 °C at 20 °C·min^−1^. The glass and melting temperatures (T_g_ and T_m_) were determined. Thermogravimetric analyses (TGA) were performed on a Setaram Setsys Evolution 16 apparatus by heating from 20 °C to 800 °C at 10 °C·min^−1^ under air atmosphere. Dynamic mechanical analyses (DMA) were performed on a TA Instrument Q800 analyzer in tension film mode equipped with a gas-cooling accessory. The characterization was carried out on samples having dimensions of 20 mm length, 5–8 mm width and a thickness of 0.2 mm. Samples were clamped and a strain of 0.04% was applied with a frequency of 1 Hz. They were firstly equilibrated at −40 °C for 5 min then heated up to 100 °C with a ramp of 3 °C·min^−1^. Finally, they were held at 100 °C for 10 min and cooled down to 20 °C. All tests were done under air. The glass transition temperature was determined from the maximum value of the tan δ peak. The crosslink density was estimated from the rubbery plateau storage modulus at Tg + 40 °C according to the theory of rubber elasticity [48] (Equation (1)):(1)$$\nu e\;=\;\frac{E\prime }{3\mathrm{RT}}$$
where  E ′ is the rubbery plateau storage modulus at T_g_ + 40 °C, R is the gas constant, T is the temperature in K corresponding to stargaze modulus at T_g_ + 40 °C. Tensile measurements (mechanical testing) were performed at room temperature using an Instron 5965 apparatus equipped with a 20 mN sensor at a crosshead displacement rate of 1 mm·min^−1^.

*Antibacterial Activity Against E. coli and S. aureus*: The anti-adherence activity of the co-networks was evaluated using two pathogenic bacterial strains: *Staphylococcus aureus* ATCC6538 (Gram-positive) and *Escherichia coli* ATCC25922 (Gram-negative), which were grown aerobically at 37 °C overnight on LB medium (Lysogeny Broth) before the bacterial adhesion tests. The samples of the networks were immersed for 1 h at 37 °C and stirred in the bacterial suspension of the two different strains (OD_600nm_ = 0.05). For *S. aureus*, OD_600nm_ = 0.05 corresponds to 153 × 10^6^ bacteria and for *E. coli*, OD_600nm_ = 0.05 corresponds to 99 × 10^6^ bacteria. The non-adherent bacteria were then removed from the network surface by 7 washes with a physiological saline buffer. The networks were then submitted to vortex and ultrasounds in a minimum volume of saline buffer to unhook fasten bacteria. Finally, 100 µL of this resulting viable bacteria suspension were inoculated onto the surface of a plate count agar Petri dish. After 24 h of incubation at 37 °C in aerobic conditions, the antibacterial properties were measured by counting CFUs for the different strains. Each experiment was performed in six replicates to allow a significant statistical analysis. The software R was used to analyze the data. The analysis of variance (ANOVA) statistical test was used and significant differences (*p* < 0.05) among antibacterial properties of the networks were detected thanks to the multiple range test of Duncan.

## 3. Results and Discussion

### 3.1. Synthesis of α,ω-Diallylic PHBHV Oligomers

Difunctional α,ω-diallylic PHBHV oligomers were synthesized in two steps (Scheme 1). The α,ω-dihydroxylic PHBHV oligomers were first obtained by a direct transesterification reaction in the presence of ethylene glycol under microwave irradiation as described in our previous study [49]. As attempted, the highly efficient heating process allows the synthesis of well-defined oligomers with very short reaction times (1 h) compared to conventional thermal processes (24 h). The structure of the resulting oligomers was confirmed by ^1^H NMR in DMSO-d6 at 70 °C (Figure 1). The molar mass was calculated by the integration of the ^1^H NMR signal at 5.25 ppm (3-3’, OCHRCH_2_ on the PHA backbone) against the signal at 3.55 ppm (B, OCH_2_CH_2_OH). Then, α,ω-diallylic PHBHV oligomers were obtained by condensation reaction [50,51] and characterized by ^1^H NMR in CDCl_3_ (Figure 2). The appearance of new signals at 5.8 ppm (H_3_, CH_2_=CHC) due to the allylic protons and 4.25 ppm (protons 6 and 7 of ethylene glycol terminal unit) confirmed the successful functionalization. Size exclusion chromatography was used to determine the polydispersity and the molar mass of the synthesized oligomers (Table 1). The value of the dispersity Ð around 1.5 shows the moderated homogeneity of the oligomers. Moreover, the oligomers are semi-crystalline and present relatively high temperature of 5% weight loss (upper of 240 °C) just like the starting polymer. It can be noticed that the value of the glass transition temperature decreases sharply when switching from native polymer to oligomers, particularly in the case of the α,ω-dihydroxylic PHBHV oligomers. This can be explained by the increase in the free volume due to the decrease of the molar masses compared to the initial PHBHV. This temperature increases again after allylation because of hydrogen bonds provided by the urethane functions.

### 3.2. Preparation and Characterization of the PHBHV-L Co-Networks 

PHBHV–linalool co-networks were synthesized via a thiol-ene reaction between PHBHV oligomers, linalool and a thiol-derived crosslinking agent (tri or tetraSH) using DMPA as a Type I photo-initiator. The well-defined oligomers were associated with an increasing percentage of linalool (0 to 100%) and the initial stoichiometry C=C/SH was fixed to 1/1. After 600 s of irradiation, materials were obtained (Figure 3).

It can be assumed that the introduction of an increasing percentage of linalool into the formulation lead to an increase of the transparency attesting of the improvement of the compatibility of the two co-networks. The conversion of C=C groups from both diallyl PHBHV oligomers and linalool was followed by FTIR in laminated conditions (Figure 4). The conversion of linalool double bonds and PHBHV oligomers depends on the functionality of thiol (Table 2). Indeed, the conversion of double bonds is total in the case of the use of TriSH showing a good equi-reactivity of double bonds. In the case of TetraSH the conversion is lower (70%). Indeed, the network quickly becomes more constrained in the presence of a crosslinking agent whose functionality is equal to 4. These results are in line with the values of the rate of polymerization, which are lower in the case of TetraSH-based co-networks (between 12 and 14 s^−1^ in the case of TriSH and between 2 and 3 s^−1^ in the case of TetraSH). This leads to a low diffusion of the radicals in the material upon light exposure and therefore a lower conversion. The water sensitivity of the networks was studied by titration by measuring the amount of water taken up after 24 h of exposure to a 75% relative humidity atmosphere. As expected, the co-networks from aliphatic PHAs and linalool are non-water-sensitive materials. A slight decrease in the co-network thickness due to the lower proportion of PHBHV in the formulation can be noticed (Table 3). The percentages of extract and water uptake are slightly higher for TriSH networks. This trend could be explained by the incomplete conversion measured in laminar conditions. It is reasonable to think that the resulting network is more heterogeneous, which makes it easier for water to enter the network that explain the increase of water sensitivity values. The introduction of an increasing proportion of PHBHV affects the hydrophobicity of the co-networks since it led to an increase in the contact angle.

The 20–80 (results not shown) and 30–70 PHBHV-L co-networks were based on TriSH being highly sticky and soft; their mechanical characterization could not be carried out. Thus, only the 50–50 PHBHV-L co-networks will be retained in the continuation of this study because of their easier handling. The investigated thermo-mechanical properties of the co-networks performed in four replicates are summarized in the Table 4. These mechanical properties allowed the determination of the Young modulus (E), the tensile strength (σ_r_) and the elongation at break (ε_r_) (Figure 5). In the case of the TetraSH-based co-networks, the reticulation with linalool has significantly modified the thermomechanical properties of the PHAs. The tensile strength decreases from 30 MPa (native PHBHV) to about 3–6 MPa. This trend is accompanied by an increase in the elongation at break from 7% (native PHBHV) to about 40% in the case of PHBHV-L networks, which is reflected in a better flexibility of the networks compare to native PHBHV polymers [51]. Furthermore, the increase of the weight fraction of PHBHV enhanced Young’s modulus from 2.9 to 5.8.

If we compare the effects of the two crosslinking agents, the result reveals an improvement of the deformability with the TetraSH. DMA analysis was performed to study the influence of linalool on the thermomechanical properties of PHBHV networks (Figure 6). It has been observed that the presence of linalool induces a decrease in the elastic modulus (E’), which can be considered as an effective plasticizing effect (Table 4). This agrees with the increase in elongation at break described above. The curve of the dissipation factor, tan δ, shows a monomodal profile, which demonstrated that the materials are homogeneous except the 50–50 PHBHV-L TriSH that seems less defined as it was attested by the very large Lα (84 °C) and width at half height at tan δ peak. It can be explained by the higher rate of reticulation that induces an increase in the cross-linking density and ν_e_ (average number of crosslinks per unit volume) in the presence of TriSH. The statistic heat-resistance temperature (T_s_) corresponding to the thermal stability of the co-networks was determined using the Equation (2):(2)$$T_{s}=0.49\;\times \left[T_{5\%}+0.6\;\times \left(T_{30\%}-T_{5\%}\right)\right].$$

It can be noticed that Ts values remain close to each other. It is important to note that this value is higher than native PHBHV (Ts = 126 °C [52]). This result is a consequence of the crosslinking, which induces the higher thermal stability of the PHBHV-L materials.

### 3.3. Bacterial Adherence

The anti-adherence properties of the co-networks were investigated using two pathogenic bacterial strains—*S. aureus* (Gram-positive) and *E. coli* (Gram-negative)—which are known to be responsible for hospital-acquired infection. Figure 7 displays the number of viable adhered bacteria on the material surface (CFU/cm^2^). The nature of the crosslinking agent has no influence on the bacterial adherence since there is no statistically significant difference between the co-networks based on Tri or Tetra-SH. Whatever the bacteria strains, the same trend is observed. It should be noticed that the CFU number of *E. coli* on the surface of the photoinduced materials after 24 h of incubation is higher than that observed for *S. aureus*. This latter result could be explained by the membrane structures of both bacterial strains. Indeed, the surfaces of *E. coli*, Gram-negative strains, which possess a relatively thin layer of peptidoglycans, are much more hydrophobic than those of *S. aureus* ones and, interestingly, promote their contact on the hydrophobic surface [53,54]. In the case of *E. coli*, a reduction of 69% or 56% in bacterial adhesion was observed when the proportion of linalool in the co-networks varied from 0 to 70 wt% for the Tri or Tetra-based co-networks, respectively. Likewise, for the *S. aureus* strain, the bacterial adhesion decreased from 78% to 87% when the percentage of linalool increases from 0 to 70 wt% for the Tri or Tetra-based co-networks, respectively. The anti-adhesive properties of the photoinduced materials are likely due to the presence of chemically grafted linalool as previously described for essential oils [26].

## 4. Conclusions

The efficient and easily performed thiol-ene reaction was used to designed bio-based antibacterial co-networks in line with green chemistry. New sustainable materials based on a bio-compatible PHBHV and GRAS compound, i.e., linalool, have been synthesized by varying the linalool proportion and the nature of the crosslinking agent. On the one hand, the use of the TetraSH and the higher PHBHV proportion led to an increase in the thermo-mechanical performance. These materials obtained from PHAs exhibit significantly higher elasticity properties in comparison with those of native PHAs. Furthermore, the synergic association with the active terpene, linalool, has made it possible to functionalize the PHAs to develop a bioactive material with an improvement of the antibacterial properties whatever the cross-linker agent and the bacterial strains (around 63% for *E. coli* and 82% for *S. aureus* strains). Moreover, the impact of linalool on the bacterial adhesion is higher in the case of *S. aureus*. Thereby, these new renewable co-networks represent a very interesting antibacterial material, in which the thermo-mechanical properties can be tuned by varying the PHBHV proportion. Finally, the variety and richness of essential oils rich in terpene derivatives offer a wide range of possibilities, since this process can be extended to other terpenes or mixtures in order to exacerbate the antibacterial but also the mechanical properties of the materials thus synthesized.

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
