# Peer review of "Co-Networks Poly(hydroxyalkanoates)-Terpenes to Enhance Antibacterial Properties"

_bioengineering, 2020, doi:10.3390/bioengineering7010013_

Round 1

Reviewer 1 Report

Modjinou et al. synthesized antibacterial PHAs-Linalool co-networks via thiol-ene click reaction with high mechanical resistance and flexibility. In introduction part, they represent material properties and reaction details; however, they do not connect what they developed and what they envision with those materials. For example, why we need novel anti-adherent materials with high mechanical resistance and flexibility and what kind of applications they target. Good transition is needed in both introduction and conclusion. 

Secondly, line 28-30, Modjinou and co-workers state the issue of antibiotic resistance which was supported by references #2-3, written in French language. Bioengineering is a universal journal and papers are published there must be in a language that everybody can understand. Antibiotic resistance is a very well-known global health problem and it is reported by several universal scientific resources including papers and global reports. These citations must be updated. 

In materials/methods section, authors described reactions and methods clearly. They only need to indicate how many cells (CFU/ml) corresponds to OD600 = 0.05 (Antibacterial activity section (line 177).

In 3.1, line 200, authors claim that dispersity around 1.5-1.6 shows the homogeneity of the oligomers. That is not right. 1.5 - 1.6 dispersity cannot be classified as low dispersity. There are couple of studies already showed synthetically developed unimolecular oligomers. They should attest 1.5 of dispersity as moderate dispersity. 

In 3.3 Bacterial adhesion section, authors claim increasing the linalool content in co-networks led a decrease in E. coli and S. aureus adhesion. However, in the case of 50-50 PHBHV-L,  E. Coli exhibits an increase in the adhesion compared to reference sample having 0% Linalool. How authors explain this? Is there any data for 40-60 PHBHV-L? This data can be the key to explain this trend. Maybe there is a optimal concentration of linalool needed to initiate potent anti-adherent properties and this is hard to judge based on two co-networks synthesized. More data is needed to present. 

Authors investigated anti-adherent properties of co-network based on cell cultures and colony counting methods which are the standard methods to demonstrate adhesion of bacteria on the surface. These methods involve ultra-sonication and vortex to remove adherent bacterial cells on the surface. Even though these techniques are acceptable and powerful to remove attached bacteria, some bacteria may still persist to stay on the surface. As a complementary technique to colony counting, authors need to consider Live/dead assay and imaging with confocal microscopy to demonstrate the surfaces after cell cultures. By this way, they will prove that these materials are biocidal, supporting their hypothesis on lines 308-310.

Authors also mentioned degradability of these materials at some parts. There is no data indicated about biodegradability and antibacterial properties of degraded products.  If they do not represent data on biodegradability, they need to remove this feature from the paper or support it with data.

Typo in Conclusion part, line 317 must be the efficient and easily performed (or performable) thiol-ene click reaction was used to design...

Author Response

Dear Editor,

We thank the reviewers for their comments that help us to improve the quality of our manuscript. We include point by point responses to their comments. We proposed here a revised version for publication in the special issue ‘ESBP2019’.

Sincerely yours,

Professor Valérie Langlois

 Reviewer 1

Modjinou et al. synthesized antibacterial PHAs-Linalool co-networks via thiol-ene click reaction with high mechanical resistance and flexibility. In introduction part, they represent material properties and reaction details; however, they do not connect what they developed and what they envision with those materials. For example, why we need novel anti-adherent materials with high mechanical resistance and flexibility and what kind of applications they target. Good transition is needed in both introduction and conclusion. 

We have taken into consideration the reviewer's remark and clarified our strategy in the manuscript and we have specified the potential interest of these new PHA-based biomaterials. There are many studies on antibacterial coatings, involving different mechanisms of action. However, our approach differs from this work because our objective is to modify the mechanical properties of PHAs by using simple chemistry and at the same time to functionalize these materials by conferring an antibacterial activity. This approach differs from the studies described in the literature using terpenes as antibacterial agents that act by diffusion of terpene molecules.

PHAS are biopolymers with clearly established biocompatibility properties. Thus, it is possible to envisage materials that could be used in the biomedical field. PHAs scl are not yet widely used in the biomedical field due to their limited mechanical properties. mcl PHAs are often better suited but these polymers are not commercially available. Our approach results in a material with mechanical properties similar to mcl PHAs. The material properties could be modulated according to the chemical composition of the network. The incorporation of terpene units, inducing anti-adhesive activity against pathogenic bacteria, could be a major advantage to widen the potential applications of PHAs.

Secondly, line 28-30, Modjinou and co-workers state the issue of antibiotic resistance which was supported by references #2-3, written in French language. Bioengineering is a universal journal and papers are published there must be in a language that everybody can understand. Antibiotic resistance is a very well-known global health problem and it is reported by several universal scientific resources including papers and global reports. These citations must be updated. 

We modified the references (1-5).

In materials/methods section, authors described reactions and methods clearly. They only need to indicate how many cells (CFU/ml) corresponds to OD600 = 0.05 (Antibacterial activity section (line 177).

For S. aureus, OD600 = 0.05 corresponds to 153 x 10E6 bacteria

For E. coli, OD600 = 0.05 corresponds to 99 x 10E6 bacteria

We precised the experimental part.

In 3.1, line 200, authors claim that dispersity around 1.5-1.6 shows the homogeneity of the oligomers. That is not right. 1.5 - 1.6 dispersity cannot be classified as low dispersity. There are couple of studies already showed synthetically developed unimolecular oligomers. They should attest 1.5 of dispersity as moderate dispersity. 

The dispersity of the initial PHBHV is 2.6. Therefore the oligomers obtained have a lower dispersity (1.5/1.6) because they are purified by precipitation after reaction. We introduced the term moderated as it was suggested by the reviewer.

In 3.3 Bacterial adhesion section, authors claim increasing the linalool content in co-networks led a decrease in E. coli and S. aureus adhesion. However, in the case of 50-50 PHBHV-L,  E. Coli exhibits an increase in the adhesion compared to reference sample having 0% Linalool. How authors explain this? Is there any data for 40-60 PHBHV-L? This data can be the key to explain this trend. Maybe there is a optimal concentration of linalool needed to initiate potent anti-adherent properties and this is hard to judge based on two co-networks synthesized. More data is needed to present. 

Bacterial section has been modified accordingly. We remove the ambiguous sentence: “The introduction of an increasing proportion of linalool led to a statistically significant decrease of the bacterial adhesion on the material surface (p < 0.001) (Figure 6).” Indeed, the number of CFU for 50/50 (PHBHV-co-L) is the same than that observed with 100/0 (PHBHV-co-L) for E. coli.

We added the following sentences: “In the case of E. coli strain, the reduction of bacterial adhesion requires a percentage of at least 70% by weight of linalool whereas in the case of S. aureus, a decrease was observed from 50%. It should be noticed that the CFU number of E. coli on the surface of the photoinduced materials after 24h of incubation is higher than that observed with S. aureus. This later result could be explained by the membrane structures of both bacterial strains. Indeed, the surfaces of E. coli, gram-negative strains, which possess a relatively thin layer of peptidoglycans, are much more hydrophobic than those of S. aureus ones and, interestingly favour their contact on hydrophobic surface.

The idea is to introduce the maximum of linalool in the photoinduced materials to observe an anti-adhesive effect. It must be pointed out that 70% is the maximum weight value of linalool that it could be introduced in the photoinduced materials to obtain a handling material. By increasing the linalool weight mass (> 70 wt%), materials are too sticky and cannot be handled. It is also interesting to notice that samples with TriSH are very brittle. Therefore, we don’t think that additional experiments should be made with 40-60 PHBHV-L as we need to have the maximum of linalool within the materials to induce anti-adherence effects.

Authors investigated anti-adherent properties of co-network based on cell cultures and colony counting methods which are the standard methods to demonstrate adhesion of bacteria on the surface. These methods involve ultra-sonication and vortex to remove adherent bacterial cells on the surface. Even though these techniques are acceptable and powerful to remove attached bacteria, some bacteria may still persist to stay on the surface. As a complementary technique to colony counting, authors need to consider Live/dead assay and imaging with confocal microscopy to demonstrate the surfaces after cell cultures. By this way, they will prove that these materials are biocidal, supporting their hypothesis on lines 308-310.

The reviewer is right. We only demonstrate the anti-adhesive effect of linalool when introduced within the photoinduced materials. Therefore, we remove all the sentences related to the biocide properties of our materials. Only the anti-adhesion properties of the surface of our materials are observed.

To demonstrate the survival and the death of bacteria, the use of propiodium iodide and FM@95 could be useful. Indeed, the general procedure for acquiring fluorescence images have been previously described: before the fluorescence images were done, samples should be stained with 30 μM of PI, and 50 μg mL−1 of FM®5-95 dye. The PI and FM®5-95 stains were sequentially excited at 514 nm; and their fluorescence emissions should be collected in the respective range 650-800 nm and 570-615 nm. However, two main problems have emerged: 1) the materials are not fully transparent so that the observation of bacteria cells remains difficult by confocal laser scanning microscopy and 2) the surface of the materials containing linalool are so sticky that the removal of dyes from the surface is difficult. Electrostatic adhesion is observed at the surface of the materials.

After consideration of all these concerns, we focus only to the colony counting technique to evaluate the anti-adherence properties of our materials.

Authors also mentioned degradability of these materials at some parts. There is no data indicated about biodegradability and antibacterial properties of degraded products.  If they do not represent data on biodegradability y, they need to remove this feature from the paper or support it with data.

We have just mentioned that PHAs are biodegradable and bio-sourced polyester ( ref 32. Brandl, H.; Gross, R. A.; Lenz, R. W.; Fuller, R. C. Plastics from Bacteria and for Bacteria: Poly(Beta-Hydroxyalkanoates) as Natural, Biocompatible, and Biodegradable Polyesters. Adv. Biochem. Eng. Biotechnol. 1990, 41, 77–93).

Typo in Conclusion part, line 317 must be the efficient and easily performed (or performable) thiol-ene click reaction was used to design...

We modified as it was suggested by the reviewer.

Reviewer 2 Report

Comments: This is a nice manuscript with a clear concept. PHA-terpene based materials were synthesized and characterized properly. The goal to obtain material with tunable thermo-mechanical properties and at the same time having antibacterial activity was achieved. I recommend to accept the manuscript after minor revisions as noted below:

1.English language and style should be improved. The mistakes are hard to be missed. Missing comas are found throughout the text (this makes it difficult to understand the sentence). The sentences are composed poorly that it's hard to read. Words are poorly chosen. Articles are used incorrectly. Some examples of the language problems: 

page 1 line 30: 'going to be stopped' should be 'going to stop'

page 1 line 30-31: 'to design of real' should be 'to design a well'

page 1 line 32: 'prevents the bacterial' should be 'prevents bacterial'

page 1 line 37: 'compounds as N-halamines' should be ' compounds such as N-halamines'

2. Many citations are missing. Some examples on page 1: references for 'hospital infections are due to...' 'classification of antibacterial materials into two types' 'copper' 'halogens' 'essential oils' are missing

3. Introduction: the authors mentioned that one of the reasons for using bio-based polymer is that they tried to promote bio-resources as an alternative for petroleum sources. Petroleum has nothing to do with antibacterial materials.

4. In Introduction, authors mentioned that there are 2 types of PHA: short chain and medium chain. How many repeat units contained in short and medium type? And which PHA type is used in their work? (it's found that authors used the short chain type after reading the discussion section). This should be explained in the earlier part of the manuscript. 

5. PHA and PHBHV are two different things where PHBHV is a PHA-based copolymer. Now the problem is that the authors did not explain in the Introduction that they're going to use PHBHV. Authors jumped directly to Scheme 1 explanation, where authors suddenly mentioned PHBHV. This will be confusing for the readers.

6. Scheme 1 is hard to understand at the first glance: a) it's better to put the name of each chemical structure in the scheme so readers can refer to it while reading the text. b) The structure name should be placed close to the correlated structure drawing, otherwise it'll be confusing. For example, the text 'dihydroxyclic PHA' is located exactly in between the structure of PHBHV before and after transesterification. So it's hard to tell which structure drawing that the text 'dihydroxyclic PHA' refers to. c) The first step of the synthesis is the reaction between PHBHV and ethylene glycol. However, the chemical structure shown in Scheme 1 is the structure of PHA. Perhaps the scheme is a simplified representative of the reaction? So I would like to suggest the authors to make it clear for the readers.

7. Please include the FTIR spectra in the manuscript.

8. How the contact angle was measured? Is it a static or dynamic measurement? What equation was used to determine the contact angle? If it's static angle, it's known that static angle has no physical meaning. So authors should re-measure the contact angle using dynamic measurement.

9. Authors stated that the thickness was decreased as the PHBHV proportion was lower. This is not true if readers refer to Table 3. For example, 30% PHBHV thickness was 0.28 mm, and 50% PHBHV was 0.16 mm. Also, please explain the effect of the co-network composition on the obtained thickness. 

10. What is the purpose of measuring the 'soluble extract'? In the Materials and Methods section, it's stated that extraction was done to remove unreacted monomer using the appropriate solvent (i.e., DCM for PHBHV and ethyl acetate for linalool). Removing unreacted monomer is basic requirement in preparing any material. So please explain why the soluble extract data is important? Also, in the text, authors stated that 'water entered the network and extracted the soluble part'...typo, it should be DCM and ethyl acetate.

11. Please switch the position of Table 3 and Figure 4 because explanation about Table 3 comes first in the manuscript.

12. Linalool is an active antibacterial, and in the text authors explained how linalool disrupts the bacterial cell wall. On the other hand, authors highlighted the anti-adherence properties of the materials. Active antibacterial material (i.e., having killing activity) is different from passive antibacterial material (i.e., having repelling property). Please make it clear.

Author Response

Dear Editor,

We thank the reviewers for their comments that help us to improve the quality of our manuscript. We include point by point responses to their comments. We proposed here a revised version for publication in the special issue ‘ESBP2019’.

Sincerely yours,

Professor Valérie Langlois

 Reviewer 2

1.English language and style should be improved. The mistakes are hard to be missed. Missing comas are found throughout the text (this makes it difficult to understand the sentence). The sentences are composed poorly that it's hard to read. Words are poorly chosen. Articles are used incorrectly. Some examples of the language problems: 

page 1 line 30: 'going to be stopped' should be 'going to stop'

page 1 line 30-31: 'to design of real' should be 'to design a well'

page 1 line 32: 'prevents the bacterial' should be 'prevents bacterial'

page 1 line 37: 'compounds as N-halamines' should be ' compounds such as N-halamines'

We corrected and modified our manuscript.

Many citations are missing. Some examples on page 1: references for 'hospital infections are due to...' 'classification of antibacterial materials into two types' 'copper' 'halogens' 'essential oils' are missing

We modified the manuscript and added the following references:

References:

R. S. Evans, R. H. Abouzelof, C. W. Taylor, V. Anderson, S. Sumner, S. Soutter, R. Kleckner and J. F. Lloyd, AMIA Annual Symposium proceedings. AMIA Symposium, 2009, 2009, 178-182. CDC, MMWR Morb. Mortal. Wkly. Rep. , 1992, 41, 783-787. R. M. Klevens, J. R. Edwards, C. L. Richards, Jr., T. C. Horan, R. P. Gaynes, D. A. Pollock and D. M. Cardo, Public health reports (Washington, D.C. : 1974), 2007, 122, 160-166. S. S. Magill, J. R. Edwards, W. Bamberg, Z. G. Beldavs, G. Dumyati, M. A. Kainer, R. Lynfield, M. Maloney, L. McAllister-Hollod, J. Nadle, S. M. Ray, D. L. Thompson, L. E. Wilson and S. K. Fridkin, New Engl. J. Med., 2014, 370, 1198-1208. P. W. Stone, M. Pogorzelska-Maziarz, C. T. Herzig, W. L. M.;, E. Y. Furuya, A. Dick and E. Larson, Am. J. Infect. Control., 2014, 42, 94-99. Introduction: the authors mentioned that one of the reasons for using bio-based polymer is that they tried to promote bio-resources as an alternative for petroleum sources. Petroleum has nothing to do with antibacterial materials.

Indeed, the bio-sourced nature is not related to the antibacterial property. We have chosen to highlight the bio-sourced nature of the terpenes responsible for the antibacterial activity of the material. The association with biopolymers, PHAs, allows us to develop bio-sourced materials that are consistent with today's new socio-economic issues. We have modified our manuscript to improve understanding.

In Introduction, authors mentioned that there are 2 types of PHA: short chain and medium chain. How many repeat units contained in short and medium type? And which PHA type is used in their work? (it's found that authors used the short chain type after reading the discussion section). This should be explained in the earlier part of the manuscript. 

We added the following sentences:Among the bio-based polymers, poly(hydroxyalkanoate)s (PHAs) are aliphatic polyesters produced by bioconversion as intracellular nutriment storage materials inside bacteria [29-31]. Two type of PHAs with various physical properties can be distinguished according to the length of the side chains: short chain length (scl-PHAs) possessing alkyl side chains having up to two carbon atoms and medium chain length PHAs, or mcl-PHAs, having at least three carbon atoms on their side chains. Scl-PHAs are semi-crystalline, rigid and brittle and on the opposite, medium chain length (mcl-PHAs) present elastomeric properties. Finally, according to their promising properties such as biodegradability and biocompatibility, they can be used in biomedical applications [32]

PHA and PHBHV are two different things where PHBHV is a PHA-based copolymer. Now the problem is that the authors did not explain in the Introduction that they're going to use PHBHV. Authors jumped directly to Scheme 1 explanation, where authors suddenly mentioned PHBHV. This will be confusing for the readers.

It was clarified in the introduction. PHA is the generic term for bacterial and PHBHV is a copolymer and is one of the most common and commercially available PHAs.

Scheme 1 is hard to understand at the first glance: a) it's better to put the name of each chemical structure in the scheme so readers can refer to it while reading the text. b) The structure name should be placed close to the correlated structure drawing, otherwise it'll be confusing. For example, the text 'dihydroxyclic PHA' is located exactly in between the structure of PHBHV before and after transesterification. So it's hard to tell which structure drawing that the text 'dihydroxyclic PHA' refers to. c) The first step of the synthesis is the reaction between PHBHV and ethylene glycol. However, the chemical structure shown in Scheme 1 is the structure of PHA. Perhaps the scheme is a simplified representative of the reaction? So I would like to suggest the authors to make it clear for the readers.

A preliminary functionalization of poly(hydroxybutyrate-co-hydroxyvalerate) PHBHV is required to introduce the unsaturated terminal groups on PHBHV (Scheme 1).  The a,ω-dihydroxylic PHBHV oligomers are first synthesized by a transesterification in presence of ethylene glycol which further reacted with allyl isocyanate to obtain α,ω-diallylic PHBHV. The influence of the nature and the content of the crosslinking agent based on multifunctional thiol (TriSH or TetraSH) on the thermo-mechanical performances of the co-networks were carefully evaluated.

We clarified scheme 1.

Please include the FTIR spectra in the manuscript.

The FTIR spectra is included (Figure 4).

How the contact angle was measured? Is it a static or dynamic measurement? What equation was used to determine the contact angle? If it's static angle, it's known that static angle has no physical meaning. So authors should re-measure the contact angle using dynamic measurement.

The contact angle was measured by static measurement. The values are compared between the different samples and the difference is significant with a deviation of 17°. For this reason we propose to keep these values.

Authors stated that the thickness was decreased as the PHBHV proportion was lower. This is not true if readers refer to Table 3. For example, 30% PHBHV thickness was 0.28 mm, and 50% PHBHV was 0.16 mm. Also, please explain the effect of the co-network composition on the obtained thickness. 

The dispersion of the measurements is 0.07 which is very important compared to the values of the measured thickness. Therefore, we prefer to remove the thickness measurement which does not provide any relevant information.

What is the purpose of measuring the 'soluble extract'? In the Materials and Methods section, it's stated that extraction was done to remove unreacted monomer using the appropriate solvent (i.e., DCM for PHBHV and ethyl acetate for linalool). Removing unreacted monomer is basic requirement in preparing any material. So please explain why the soluble extract data is important? Also, in the text, authors stated that 'water entered the network and extracted the soluble part'...typo, it should be DCM and ethyl acetate.

We agree with the reviewer. We extracted the non-crosslinked products with dichloromethane to remove the non-crosslinked PHBHV and with ethyl acetate to remove the non-crosslinked linalool.

We clarified the following sentence page 7 “It is reasonable to think that the resulting network is more heterogeneous, which makes it easier for water to enter the network that explain the increase of water sensitivity values”.

Please switch the position of Table 3 and Figure 4 because explanation about Table 3 comes first in the manuscript.

We switched Table 3 and Figure 4.

Linalool is an active antibacterial, and in the text authors explained how linalool disrupts the bacterial cell wall. On the other hand, authors highlighted the anti-adherence properties of the materials. Active antibacterial material (i.e., having killing activity) is different from passive antibacterial material (i.e., having repelling property). Please make it clear.

Free linalool is anti-bacterial and the mechanism of antibacterial activity is described in the literature In our study linalool is immobilized within the network, it cannot diffuse as shown by the results of the extractible contents. We agree with the reviewer concerning the mode of action of immobilized linalool which cannot be similar to free linalool. The observed results are most likely related to a mechanism of antiadhesion of bacteria on the material and not to biocidal activity.Even if it is difficult to suggest the same mechanism, we observe the same trend between positive and negative grams between cross-linked and free linalool. This can also be explained by the different structure of gram positive such as Aureus bacteria walls, which are thick but with a simple chemical structure. The walls of gram-negative bacteria are thinner but with a much more complex chemical structure.

We modified our manuscript.
